# Professional Competence and Its Effect on the Implementation of Healthcare 4.0 Technologies: Scoping Review and Future Research Directions

**DOI:** 10.3390/ijerph20010478

**Published:** 2022-12-28

**Authors:** Abey Jose, Guilherme L. Tortorella, Roberto Vassolo, Maneesh Kumar, Alejandro F. Mac Cawley

**Affiliations:** 1Department of Industrial and Systems Engineering, Pontificia Universidad Católica de Chile, Santiago 7820000, Chile; 2Department of Mechanical Engineering, The University of Melbourne, Melbourne, VIC 3010, Australia; 3IAE Business School, Universidad Austral, Buenos Aires B1630FHB, Argentina; 4Department of Production and Systems Engineering, Universidade Federal de Santa Catarina, Florianopolis 88040-900, Brazil; 5Logistics and Operations Management Section, Cardiff Business School, Cardiff University, Cardiff CF10 3EU, UK

**Keywords:** Healthcare 4.0, scoping review, healthcare technology, competence, digital health

## Abstract

Background: The implementation of Healthcare 4.0 technologies faces a number of barriers that have been increasingly discussed in the literature. One of the barriers presented is the lack of professionals trained in the required competencies. Such competencies can be technical, methodological, social, and personal, contributing to healthcare professionals managing and adapting to technological changes. This study aims to analyse the previous research related to the competence requirements when adopting Healthcare 4.0 technologies. Methods: To achieve our goal, we followed the standard procedure for scoping reviews. We performed a search in the most important databases and retrieved 4976 (2011–present) publications from all the databases. After removing duplicates and performing further screening processes, we ended up with 121 articles, from which 51 were selected following an in-depth analysis to compose the final publication portfolio. Results: Our results show that the competence requirements for adopting Healthcare 4.0 are widely discussed in non-clinical implementations of Industry 4.0 (I4.0) applications. Based on the citation frequency and overall relevance score, the competence requirement for adopting applications of the Internet of Things (IoT) along with technical competence is a prominent contributor to the literature. Conclusions: Healthcare organisations are in a technological transition stage and widely incorporate various technologies. Organisations seem to prioritise technologies for ‘sensing’ and ‘communication’ applications. The requirements for competence to handle the technologies used for ‘processing’ and ‘actuation’ are not prevalent in the literature portfolio.

## 1. Introduction

The healthcare industry is in the phase of technological transformation where care providers and patients are partaking in more digitalised and virtualised activities than ever before by using various Internet communication technologies (ICTs). These technologies can transform the healthcare industry from a focused and compassionate system to a value-oriented system that can ensure proactive preventive measures with optimum resource utilisation [1], and improve the organisation’s overall performance by providing a digitally controlled management system [2]. Healthcare is the slowest of all industries to adopt digital technologies [3], as it primely engages a patient-specific human–human interaction for its services. However, industrial production generally builds on standards and norms that may mainly involve human–machine or machine–machine interactions [4]. Recently, there has been an increased academic and managerial interest in exploring the challenges in the technology adoption process in the healthcare sector. A detailed understanding of digital technologies and their boundaries may benefit the stakeholders, including healthcare providers, patients, and interested researchers, in adopting Healthcare 4.0 (H4.0) [5].

Digitalisation has elicited an increase in data generation of a different kind, which creates pressure for healthcare personnel who are expected to be competent to utilise the data as meaningful information with to their full potential [6]. The ability and knowledge required to manage these technologies is an example of how the competence of healthcare professionals is essential to ensure the system’s efficient functioning.

Since H4.0 technologies are expected to have an essential role in the healthcare delivery process, this research focuses on how the literature has studied the different competencies of healthcare professionals and their effect on adopting H4.0 ICTs in their organisations. Rather than answering a particular question, scoping reviews provide a broader overview of the evidence [7,8], allowing us to propose a research agenda and identify its implications for policy or practice.

To narrow down the definition of competencies, we base our study on the ones previously defined by Hecklau [9]: technical, methodological, social, and personal competencies. Subsequently, we affirm how the literature discusses the competence requirement for adopting Healthcare 4.0 against the technology bundle, i.e., sensing–communication (S–C) and processing–actuation (P–A) by Tortorella [10]. To determine the relevance of the literature for the given competency to adopt the corresponding ICT, we searched using the factors of (i) citation frequency, (ii) evidence impact level, and (iii) overall relevance level.

### 1.1. Background

#### 1.1.1. Healthcare 4.0: Competence Requirement

The manufacturing industry is undergoing a new industrial revolution, Industry 4.0, with the integration of new technologies, such as cloud computing, 3D-printing technology, cyber physical systems (CPSs), Internet of Things (IoT), Internet of Services (IoS), and Big data [11]. These new technologies have extended to healthcare services under Healthcare 4.0, generating a revolution of the entire healthcare value chain, directly affecting medicine and medical equipment production, hospital care, non-hospital care, healthcare logistics, and financial and social systems [12]. These technologies focus on providing personalised care and services that treat people with compassion and respect [3]. This helps the reduction in costs and ensures the comfort of all system users, particularly patients. However, the adoption process often seems chaotic when healthcare professionals are forced to take on additional responsibilities along with their routine patient care. Due to various factors, organisations typically struggle to manage change initiatives effectively, which raises the demand for a collective effort from the stakeholders to bring the necessary changes in the Healthcare 4.0 implementation policies [13,14]. In this context, the competence of workers related to the adoption and use of novel digital technologies plays a critical role in the organisation’s optimum performance after implementing Healthcare 4.0. The rapid change occurring due to the incorporation of Healthcare 4.0 demands an increase in the capacity for effective organisational action through knowledge and understanding. In particular, doctors and nurses face challenges with improving medical competence and advanced patient interactions with new technologies. This creates a requirement for handling the relationship between human resource management and Healthcare 4.0 [15].

The literature confers competency in a variety of settings. Whelan [16] defines competency as assessing an employee’s ability to perform the skills and tasks of their position as mentioned in their job description. Validating the competency of healthcare staff is indispensable to providing safe patient care. Therefore, acquiring competency is a continual process that ensures healthcare organisations offer high-quality care to their customers and patients. Locsin [17] argued that competence in healthcare is associated with performance and the quality of the individual. Castle [18] compared the confidence vs. competence of healthcare professionals’ basic life support skills. The combination of training and clinical exposure improves the confidence and competence of healthcare workers. Kak [19] provided a framework for understanding the significant factors affecting healthcare providers’ competence and measurements. Generally, competence involves the factors of knowledge, skills, abilities, and traits, which determine healthcare workers’ ability and readiness to provide high-quality services.

The shortfall of updated skills and role requirements often results in an inability to perform new functions. In such cases, organisations may arrange adequate tools to help their employees to gain competencies in the deficient areas [20]. The department managers, experienced staff members, and educators are the key personnel involved in competence development in the organisation [16], and a perceived competence gap can be reduced by arranging planned training [18,21]. Unsurprisingly, those engaged with processes need to perceive the competencies more than those who deal with positions [22]. In a healthcare organisation, managers need to be aware of the motivations of their employees in order to create an environment conducive to growth and change, and they need to consider that professionals are significant resources in implementing change effectively [23]. When there is a need to implement new technology, it greatly influences the employee’s attitudes [24].

#### 1.1.2. The Competence Framework

The challenges related to the competence of the existing workforce in the healthcare industry may need new approaches to cope with the technological changes. Adopting new technologies often causes problems due to improper human resource management. Hecklau [9] established a competence framework for adopting Industry 4.0 technologies, defining four competencies: technical, methodological, social, and personal. Under each set, there are several skills associated with it. A modified competence framework in the context of competence requirements for adopting Healthcare 4.0 is provided in Table 1. The competence requirements may differ across industries, and there is a minimum requirement for each skill for the better adoption of the technologies. The gap can be assessed by comparing the minimum requirements of competence and the available competence.

## 2. Materials and Methods

Our research methodology followed the framework that Arksey and O’Malley [25] proposed for scoping reviews, consisting of the following steps: (1) identifying the research question, (2) identifying relevant studies, (3) study selection, (4) charting the data, and (5) collating, summarising, and reporting the results.

### 2.1. Identifying the Research Question

When implementing Healthcare 4.0, several barriers can impact the organisation’s performance [26]. Additionally, the pace of implementation may depend on multiple factors, including the readiness of the workers, affordability, and availability of the new system for the organisation. Hecklau [9] identified competencies for a digitised and interconnected world and compiled a conceptual framework in the context of Industry 4.0. The competence requirements for adopting Healthcare 4.0 also need a reference point for holistic human resources management. This study evaluated the individual competence requirements discussed in the literature. We formulated four research questions to guide the scoping review:

**RQ1:** 
*Have competencies and attributes related to adopting H4.*
*0 been addressed in the literature? If so, which ones?*


**RQ2:** 
*How much has each competency been addressed in the literature, and which is more prevalent when adopting H4.0?*


**RQ3:** 
*Has the literature addressed challenges of competence development relating to adopting H4.0? If so, which ones?*


**RQ4:** 
*What are the potential research directions regarding competence requirements that facilitate the adoption of H4.0?*


### 2.2. Identifying the Relevant Studies

A critical step in a structured literature review is the use of the appropriate set of keywords to guarantee that the relevant studies related to Healthcare 4.0 and Competence are selected. The search process for the relevant studies was conducted in two stages. First, we performed a search using a limited number of keywords which allowed us to determine the relevant literature. In the second stage, we expanded the search keywords using the one obtained from the relevant literature in the first stage. This process allowed us to overcome the potential problem of the search string being overly specific or entailing (partially) misleading buzzwords.

In the initial search, the first set of keywords (Healthcare 4.0 OR Smart health OR eHealth OR mHealth OR Digital health) AND (Hospital) AND (Competenc* OR Skill OR Knowledge) combined to retrieve publications that used them in the title, abstract, and keywords. The asterisk (*) is used to truncate the keywords to find singular and plural forms of words and variant endings. Using the AND operator in the search process significantly reduced the occurrence of misleading results. We searched publications from Web of Science, PubMed, and Scopus with filtering criteria as journal articles, book chapters, and conference proceedings published in the English language after 2011 (as the term Healthcare 4.0 is derived from Industry 4.0, not formally acknowledged before 2011) [10]. However, we obtained a relatively low total number of search results of 731 (refer to Table 2).

In the second stage, we randomly selected five articles from each database to compare their keywords with those from the research dimensions used in the first stage [27]. The objective was to consider those different taxonomies associated with a given subject, potentially compromising the search. From the comparisons, we identified the need to add another combination of keywords to our search: Clinic OR Nursing home OR Health service OR Care provider OR Expertise OR Literacy OR Learn* OR Experience. Using this information, we modified the search, extending the keyword search string: (‘Healthcare 4.0’ OR ‘Smart health’ OR ‘eHealth’ OR ‘mHealth’ OR ‘Digital Health’) AND (‘Hospital’ OR ‘Clinic’ OR ‘Nursing home’ OR ‘Health service’ OR ‘Care provider’) AND (‘Competenc*’ OR ‘Skills’ OR ‘Knowledge’ OR ‘Expertise’ OR ‘Literacy’ OR ‘Learn*’ OR Experience), which provided us with much broader search results. This second search with the same filtration criteria as the first search returned 4976 publications (refer to Table 3) across the databases. Both search stages reported here were conducted between November and December 2021. Mendeley desktop software was used for the selection and sorting of the journals.

### 2.3. Study Selection

The researchers’ familiarity with H4.0 helped us to determine inclusion and exclusion criteria. As the first criteria, we selected journal articles, book chapters, and conference proceedings only in the English language published on or after 2011, which we opted as the initial filtration criteria of the search process. After removing duplicate publications from the portfolio, the total number of publications was reduced from 4976 to 4262. In the next exclusion step, we individually verified the alignment of titles of publications with the research topic, which resulted in the exclusion of 4053 publications from the portfolio. The remaining 209 articles whose titles matched the research topic underwent a screening process where we checked for the alignment of keywords and abstracts with the research topics. A total of 88 articles were excluded, resulting in 121 articles being considered in the eligibility step. We distributed 121 articles to 4 researchers; each one reviewed 35 articles as full texts. There was an overlap of five articles between the researchers, which allowed for cross-validation, reducing the chances of eliminating the valid ones. Whenever a consensus was not reached, a third author was asked to provide an opinion, which united the decision. Following this step, we obtained 51 articles that aligned with the research topic. We followed the PRISMA statement [28] to ensure the transparency for this scoping review. A detailed flowchart on the identification of the studies for the review is provided in Figure 1.

### 2.4. Charting the Data

Once the portfolio of relevant studies was finalised, we performed a content analysis. Given the purpose of our study, the articles were organised in a spreadsheet, including the following information: authors, year of publication, journal, H4.0 ICTs, specific application, associated competence requirements, and sub-competences in the context of applications in the publication. For the content analysis, we performed a three-step process. First, the 51 articles were revisited by one of the researchers who performed a descriptive analysis of the publications (year of publication and publication outlet). Then, we quantitatively assessed the latent content of the article, and we evaluated each publication’s perspective on aspects regarding the adoption of Healthcare 4.0 ICT and competence requirements. Out of 51 publications, 44 discussed integrating specific H4.0 technologies and associated competence from various healthcare settings, and the other 7 publications were not directly linked with any ICTs. A qualitative assessment of the sub-competencies related to each competence set was performed, namely, technical, methodological, social, and personal; we verified each sub-competence in the corpus of publications against the framework of Hecklau [9] and charted this in the Results Section, which answers RQ1.

To analyse the contribution of competence development in the selected literature, we characterised the research into two categories: the type of H4.0 ICTs/applications and the relevant competence requirements. To define the type of H4.0 technology presented in each study, we used the technology bundles proposed by Tortorella [10]: sensing and communication (S–C) and processing and actuation (P–A). In the case of sensing and communication (S–C) technologies, they correspond to all those digital technologies being used for capturing and communicating information. Those technologies are represented by: biomedical/digital sensors, IoT, big data, cloud computing, and remote control/monitoring. In the case of processing and actuation (P–A), we categorised all of the studies that involved ICTs that transform the data into information. These technologies can be used for decision making or to control systems, mechanisms, or software, such as 3D printing, collaborative robots, machine/deep learning, and augmented reality/simulation. To categorise competence, we used the proposed framework proposed by Hecklau [9], which includes technical competence (TC), methodological competence (MC), social competence (SC), and personal competence (PC). We conducted a qualitative assessment of the different sets of skill requirements discussed in the corpus of the literature against the framework.

The relevance level of the contributions from the literature was linked to the targeted research topic. It can be determined by (i) citation frequency and (ii) evidenced impact level reported in the literature [29]. We used the relevance level of the contribution to represent the different types of competence associated with integrating H4.0 ICTs in healthcare organisations. As per Pagliosa [30], two factors were considered to determine the relevance level: (1) the citation frequency of different competencies as contributors in adopting H4.0 ICTs, and (2) evidenced impact level reported in the literature. For the first criterion, we used the frequency score of 0–2 to describe each pairwise competence relationship; 0: when no relations were found, 1: when relationships were cited in up to one-third of the publications, and 2: when relationships were cited in more than one-third of the publications. For the second criterion, based on the literature review, two impact score values were assigned: (1) when competence requirements for the adoption of specific ICTs were not sufficiently addressed; (2) when competence requirements for the adoption of ICTs were discussed extensively. The overall relevance level of the pairwise relationship between competence requirement and adoption of H4.0 ICTs was provided by the product score assigned to each criterion, taking the values of 0, 1, 2, and 4. Overall relevance scores of 1 and 4 indicated the least and most contributions of competence requirements for adopting H4.0 ICTs, respectively. In contrast, 2 shows an intermediate situation, and 0 denotes the potential research opportunity where no publications discuss the relationship. This helps us to explore the relevance level of different attributes of competence in adopting individual H4.0 ICTs, which answers RQ2.

Finally, a qualitative analysis explicitly explored the implications of the sub-competencies linked with each competence set in the Discussion Section, followed by the identification of the main challenges of competence development while adopting H4.0, which answers RQ3.

### 2.5. Collating, Summarising, and Reporting the Results

In this step, the results were collated, summarised, and reported based on a thematic framework to produce a narrative account of the publication portfolio available. We performed an additional analysis to improve the consistency of the study. First, we performed a descriptive thematic analysis to collate and summarise the results. Second, based on the reported results, we developed a detailed analysis of the characteristics, contributions, and competence requirements for adopting H4.0 technologies. We reported this analysis in the Analysis of Results Section. Then, we discussed our results’ implications in a broader context, ensuring the scoping study methodology’s legitimacy for theory and practice. Finally, we identified the potential research agenda, and the conclusion provided the main result of the scoping review and expanded on the limitations.

## 3. Results

Table 4 presents a descriptive numerical summary of the publications’ portfolios. As expected, the studies conducted on the competence requirements for H4.0 were extremely recent, and in the past two years, there has been a significant increase in the number of publications. The portfolio has been published across various journals worldwide, and the publications are multidisciplinary. The journals with the most publications are listed in Table 5; BMC Health Services Research and the International Journal of Medical Informatics are leading journals with three publications on each of our research topics. In addition to the competence requirements for Healthcare 4.0, the portfolio also discusses the barriers to adopting digital technologies [31,32,33], change management [34,35], and the readiness and willingness of the stakeholders [36,37].

Table 6 exhibits H4.0 ICTs’ role and applications in adopting H4.0 ICTs while discussing the competence requirements categorised into two groups: (1) non-clinical applications that support the administration and management processes, and (2) clinical applications that are directly involved in patient care. Additionally, the applications were tagged under the bundle of roles of ICTs identified by Tortorella [10], i.e., sensing and communication (S–C) and processing and actuation (P–A). In the Discussion Section, we explored the implications for the challenges and solutions in competence development in adopting H4.0 ICTs. While exploring the competence requirement for H4.0, most of the papers analysed the adoption of the technologies in non-clinical applications in a healthcare organisation. Those technologies primarily belong to the sensing and communication (S–C) bundle, and it is evident that processing and actuation (P–A) applications are fewer in number. It is worth mentioning that the adoption of ICTs, such as 3D printing, and augmented reality/simulation, were not discussed along with the competence requirements. Most of the applications used at present are applications of the Internet of Things (IoT). In other words, the competence required for adopting IoT applications is widely studied in the literature, and such applications are becoming increasingly more common in the healthcare system. Associated competence for adopting collaborative robots/robotics and deep/machine learning is less frequently discussed, which reveals that the corpus of the selected literature explored the competence requirements for processing–actuation (P–A) ICTs the least.

The qualitative assessment of the different sets of competence requirements discussed in the corpus of literature has provided a general idea about how it prioritises competence requirements. Table 7, Table 8, Table 9 and Table 10 demonstrate each competence set, technical, methodological, social, and personal, respectively, and the discussions on technical competence in adopting H4.0 are equivocal, with state-of-the-art knowledge being the most significant contributor to the list, again categorised based on its type and difference in knowledge acquisition. The technical skills and process understanding are also sub-categorised based on their types. Problem-solving and decision-making factors are the two noticeable subskills discussed under methodological competence. Communication and the ability to work in a team are the main social competencies, and no literature specifically mentioned language skills. Out of the four main competencies, the requirements of personal competence were the least evident in the literature, and the motivation to learn and compliance were the two main subskills discussed the most frequently under personal competence. The respective radar charts (refer to Figure 2) visualises and helps us to understand how the selected literature addresses the importance of different skills in adopting H4.0 technologies.

Table 11, Table 12 and Table 13 present the literature’s content analysis, allowing us to understand the relevance of different streams of competence in the adoption of each ICT (t1: Biomedical/digital wireless sensors, t2: Internet of Things (IoT), t3: Big Data, t4: Cloud/fog computing, t5: Remote control/monitoring, t6: Collaborative robots/robotics, and t7: Machine/deep learning). Table 11 presents the citation frequencies for the four competence types and the seven technologies rendering 40 pairwise assessments; for the citations, we can observe that the technology with the highest number of citations for almost all competencies is t2 (Internet of Things). Table 12 presents the impact that technologies and competence levels have on adopting different ICTs, in this case, the four competencies significantly impact the adoption of technology t2. Table 13 shows the importance of competence in adopting technology by combining the citation and impact factors. The results indicate that the four competence levels significantly affect the implementation of technology t2 (Internet of Things). We determined that the combination of different competences with the respective ICTs, which had less relevance, may not indicate the lack of contribution in the literature as the technologies were relatively new to the industry, and many organisations at present are implementing them. The assessment of the competence requirements on this set-up is presented in the step that follows shortly.

## 4. Discussion

### 4.1. Principal Findings

This section summarises the principal results obtained while reviewing the relevant literature. When a healthcare organisation adapts Healthcare 4.0 technology, it performs better at all application levels [26]. However, the skills required by healthcare professionals to efficiently use ICTs can differ from conventional ones. Implementing Healthcare 4.0 requires that healthcare professionals can perform a set of skills. These skills include technical competencies to handle various ICTs; social competencies to guide and learn about ICTs; personal competencies to adapt to change; and methodological competencies to utilise ICTs in decision-making and problem-solving tasks in their day-to-day activities [9,78]. The specific H4.0 ICTs are discussed in the literature and have been classified as sensing–communication (S–C) and processing–actuation (P–A). In the literature, healthcare workers’ competence requirements are extensively discussed for operating technologies categorised under S–C, such as electronic health records (EHRs), telemedicine/remote consultations, and wearable devices. In the case of P–A technologies, the literature has not paid much attention to the relationship between the competencies and the adoption of such technologies, either because such technologies are not widely used or there is a gap in the research that is yet to be explored. We qualitatively identified the sub-skills associated with each competence factor against the framework presented by Hecklau [9].

### 4.2. Explicit Assessment of Various Competencies

To study the competencies and their effects on H4.0 adoption, we used the classification proposed by Hecklau [9], categorising the competencies into technical, methodological, social, and personal. By analysing the literature, we observed that technical competence is the most prominent set of skills required to utilise new technologies in the healthcare industry. Technical competence, out of the four categories, was the one that provided the highest number of papers and citations. This competence is associated with six subskills, including state-of-the-art knowledge, technical skills, process understanding, handling smart devices, apps, smart media, data/information-processing skills, and understanding IT security. The additional responsibility due to the introduction of Healthcare 4.0 technologies makes state-of-the-art knowledge a vital component. It is linked with the professional knowledge of individuals [35,57], knowledge to handle various technologies [44,64], system-specific tasks [42,50], knowledge acquired from experience [34,52] and interactions and meetings [34], knowledge to prioritise needs [70], knowledge of the pre-operative planning process [71], and knowledge of data protection [58]. A variety of technical skills, including handling Healthcare 4.0 and other eHealth technologies [41,44], the operation of computer systems and computer literacy [31], device testing and troubleshooting [42], and evaluation abilities [54], help to evaluate the individual’s technical skills. Process understanding is assessed by the ability to understand telemedicine and electronic health record management procedures [45], feedback processed with data [31], and aspects of hardware and software processing [70]. Additionally, handling smart devices, apps, smart media [47], data/information processing skills [56], and understanding IT security [61] are imperative yet related and often present together. It is common that healthcare professionals who can cope with new technologies do not have sufficient cyber security knowledge and are the victims of cyber attacks. Therefore, innovative training sessions must be incorporated to ensure cyber resilience [40]. Additionally, internal security audits and IT reviews conducted prior to using any new technology ensure the system’s trouble-free performance for intended use and guarantee cyber resilience [43].

The subsequent relevant competence is methodological, which is linked to eight sub-skills. Innovative approaches and mindsets promote creativity. The creative mindset is vital for an organisation’s sustainability and makes them allocate funds for innovation initiatives [43], entrepreneurial thinking [15], problem-solving, decision making, especially in choosing the right technology [70], analytical skills for handlining highly structured data in an efficient manner [42], research skills [39], and efficiency orientation [15], which are the critical professional abilities.

The concept of collaborative teamwork demands the ability to work in a team and to be compromising and cooperative with the team. Furthermore, communication and the ability to transfer knowledge determine how well knowledge exchange occurs within a team. Additionally, healthcare workers must possess intercultural skills to overcome breakdowns due to differences in cultures, norms, symbols, or representations [42]. The networking skills present across different workers enable sustaining relationships with the practitioner-corporatised hospitals that sell their services and other public and private healthcare structures inside and outside the region [46]. Finally, the best managers need to coordinate and provide adequate support for the team that lead toward common objectives, which depicts the requirements of leadership skills. We did not observe any studies explicitly discussing the need for language skills, which may be closely related to intercultural and communication skills.

When technological change occurs, it is evident that the preoccupied roles of the workers may change based on the new requirements. Therefore, the flexibility to accept newly deployed functions [43] and the motivation to learn about new technologies [35] for clinicians are crucial. These recent changes may have occurred due to the sustainable initiatives presented by the management. Hence, it is expected to possess sustainable mindsets for the workers [43] and to anticipate and manage changes; one must be able to tolerate ambiguities [59]. In addition to routine tasks, individuals may be forced to take care of additional responsibilities, revealing their ability to work under pressure [60]. Finally, compliance requirements, such as the need for licences and accreditations, the implementation of policies, and other region-specific regulatory and ethical obligations, are mandatory [44,46].

### 4.3. Competence Development: Challenges and Solutions

The readiness to adopt new technologies differs across countries with different healthcare systems, disease burdens, healthcare needs, infrastructure, and political agendas. It is always good to conduct an eHealth readiness assessment before implementing new technologies [44]. There must be a detailed description of workers’ roles at the macro and micro levels for the leadership, management, support, or operations to deal with the new technological changes [41]. The adoption of Internet and communication technologies, such as eHealth and mHealth, is generally rapid. It has outpaced the regulatory requirement and often shown that the professional competence of healthcare professionals at present is insufficient. Hence, it is the responsibility of corresponding government authorities and healthcare institutions to provide requisite training to keep the competence of technologies of the workers up to date [50].

Learning and training are two related processes that enhance individual competencies and permit healthcare professionals to perform better in their assigned job profiles. Usually, the core competencies cannot be imitated, and it is natural to infer that their development is eventually replaced and dynamically evolving [79]. When there is a common interest in pursuing the change in the existing system, it increases the willingness of the team to regularly interact and learn. Performing clinical simulations of the tasks allows the group to familiarise itself with the system and understand the intended requirements for the group of stakeholders. They eventually understand other aspects of the organization [42]. Moreover, maintaining a discussion within the team encourages creativity and the acquisition of new competencies for the team members [80]. Failure detection while implementing a system is a crucial ability acquired with various resources and training programs and promotes healthcare resilience. Employees working together at different levels encourage organisational learning, and this collaboration may help them to overcome inevitable operational failures [59]. Another aspect worth discussing is how knowledge exchange occurs across different institutions. The geographic proximity of the organisations causes knowledge spillover due to the pooled labour market and the interaction of other employees working in various organisations, which helps them to become acquainted with different approaches in the industry [70]. Generally, in a healthcare organisation, training sessions are scheduled events that aim to provide knowledge to individuals regarding a particular field/topic. Healthcare professionals generally demand continuous training for updating new technologies [44]. Providing comprehensive information about the changes and new technology to healthcare professionals can motivate them to accept the new care setup [46], which can be provided with on-the-job training and workshops by an internal or external entity. In particular, nursing professionals, the world’s largest workforce, need more practical tools to support the integration of disciplinary nursing knowledge [81].

On the other hand, a significant barrier to imposing training is the resistance received from the physicians. While nurse managers take the implementation interventions seriously, physicians do not perceive it as their responsibility or are sceptical about the outcomes [60]. Most of the associated skills in the frameworks are related and exist together, and the lack of specific skills may impact the pace of Healthcare 4.0′s implementation. Hence, successful implementation solely depends on the effective utilisation of the resources and the development of workers’ competencies. 

## 5. Research Agenda

In the previous sections, we provided the means to identify the existing theoretical gaps and spaces for further research related to competencies’ effects on the adoption of H4.0 technologies. This section addresses these gaps and proposes a research agenda.

### 5.1. Validation of the Different Types of Competence in Adopting Healthcare 4.0

The competence framework proposed by Hecklau [9] provides a structure for the research and can be used to study their effects on the adoption process of Healthcare 4.0 ICTs. From the amount of research conducted and the number of citations obtained, we observed that the applications of the Internet of things (IoT) and technical competence were the most prominent set of competence factors and technology discussed in the literature. The other three competence requirements for adopting other forms of ICT and their applications is less frequently explored in the literature. Although it may be possible that most organisations are in the early stages of adopting H4.0, and it may be too early to assess the different competence requirements. However, a concrete reason for this phenomenon is yet to be explored. A cross-sectional survey-based study and/or qualitative analysis of the interviews could be conducted with multiple respondents from several healthcare entities and will provide us with a holistic view of the competence factors required for adopting H4.0 ICTs. An extension to this research could address the possibilities of obtaining the answers to the impacts of other organisational aspects of adopting Healthcare 4.0 and the organisation’s performance outcomes. This may provide further opportunities for the researchers to produce the correct combinations of technologic-specific competencies.

### 5.2. Examine the Need of Job-Specific Competence Factors

We considered the fact that, being a user of the technologies, not all healthcare professionals will need to possess all the skills identified in the framework of Hecklau [9]. There was a need for the assessment of competence requirements for specific job profiles. Incorporating the relevant skills into the curriculum and on-the-job training will help reduce the potential competency gaps among individuals and locate the right person for the right job.

### 5.3. Explore the Impact of the Adoption of H4.0 on Organisational Learning Capabilities

The existing literature more commonly focuses on learning by training than organisational learning (OL) capabilities. This is mainly because accompanied training sessions are likely present after incorporating new technologies to provide a comprehensive overview of the change in a system. However, the impact of H4.0 technologies on OL capability development and the influence of these OL capabilities on the relationship between H4.0 and operational performance is worth studying. Such a study will allow us to study the individual, group, and organisational levels of the organisational learning process in-depth. Both academics and practitioners concerned with integrating H4.0 into the healthcare sector can benefit from the contributions and consequences of this proposal.

## 6. Conclusions

In this research, we performed a structured literature review on the competence requirements for adopting H4.0 ICTs. We searched publications in three databases and screened the relevant articles to consolidate a publication’s portfolio on the topic, following the pre-defined criteria. The results of the scoping review were explored through (i) a descriptive numerical summary and thematic analysis, (ii) a qualitative analysis of sub-competence against an established framework, and (iii) implications of the challenges and solutions in competence development processes. The research conducted on H4.0-related topics has existed for a long period of time (since 2011), and the contributions used to assess competence requirements for H4.0 adoption are recent. This is primarily discussed for non-clinical applications that existed prior to the origin of the term Healthcare 4.0. Healthcare organisations seem to choose ‘sensing–communication’ (S–C) ICTs as their first choice.

The corpus of literature reports the set of competencies, namely, technical, methodological, social, and personal, on the adoption of individual Healthcare 4.0 ICTs. It mainly concerns the requirements of technical competencies for non-clinical applications. A fraction of the literature that reports clinical applications has the role of sensing–communication (S–C). The competence assessments of healthcare professionals need to be seriously addressed, especially those regarding the most advanced technologies, specifically those dealing with processing and actuation (P–A). Regarding the various competence development aspects discussed in the literature, training sessions are the primary source of competence development. However, further studies are required to understand how learning and knowledge sharing at a healthcare organisation level can produce more significant benefits while adopting H4.0 technologies.

It is worth mentioning the few limitations that exist in our study, including its nature and our methodological choices. First, we chose studies published after 2010 because Healthcare 4.0 was derived from Industry 4.0, formally acknowledged in 2011. There were instances where the publications discussed various ICTs used in healthcare applications even prior to 2011 [82,83], and the publications on the competence requirements for the adoption of H4.0 ICTs significantly increased in the last two years (as shown in Table 3), which substantiated our choice of the search period for publications and helped us to obtain the relevant results. Second, one of the screening criteria was to review the articles’ titles and remove the irrelevant ones, which may eliminate the poor titles with relevant content. However, we observed that the titles of the selected articles in the final portfolio consistently reflected the full text. Finally, we used the competence framework proposed by Hecklau [9], which is based on the adoption of Industry 4.0. In a Healthcare 4.0 set-up, being a user of technologies, not all healthcare professionals need to possess these skills. However, the relevant participation of healthcare professionals at all levels, including physicians, nursing professionals, and other stakeholders, with adequate skill sets, are necessary to successfully implement H4.0 ICTs.

## Figures and Tables

**Figure 1 ijerph-20-00478-f001:**
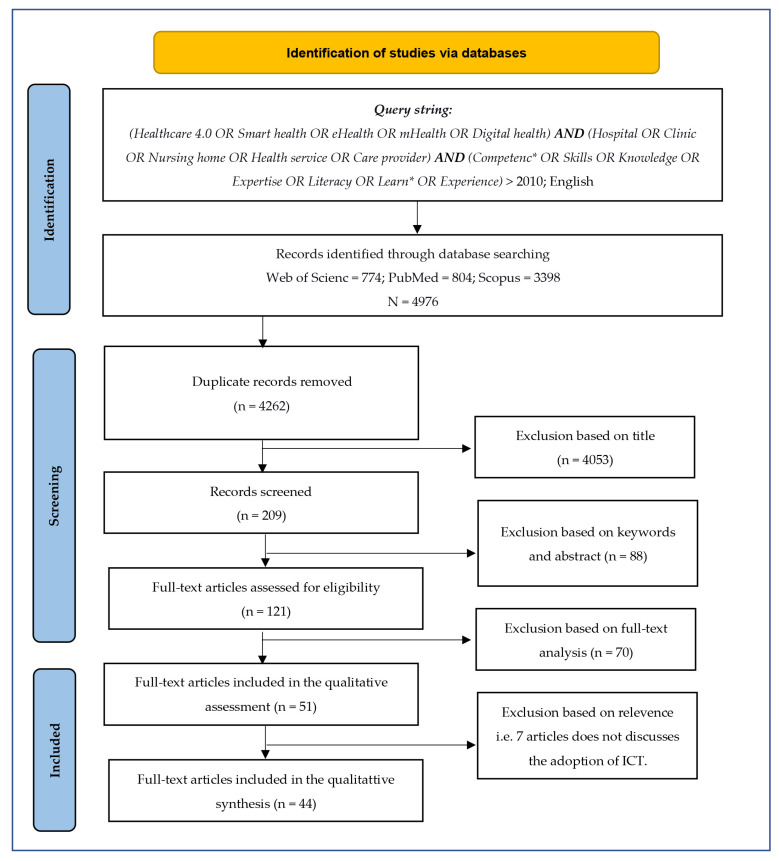
Flowchart of identification of studies via database search.

**Figure 2 ijerph-20-00478-f002:**
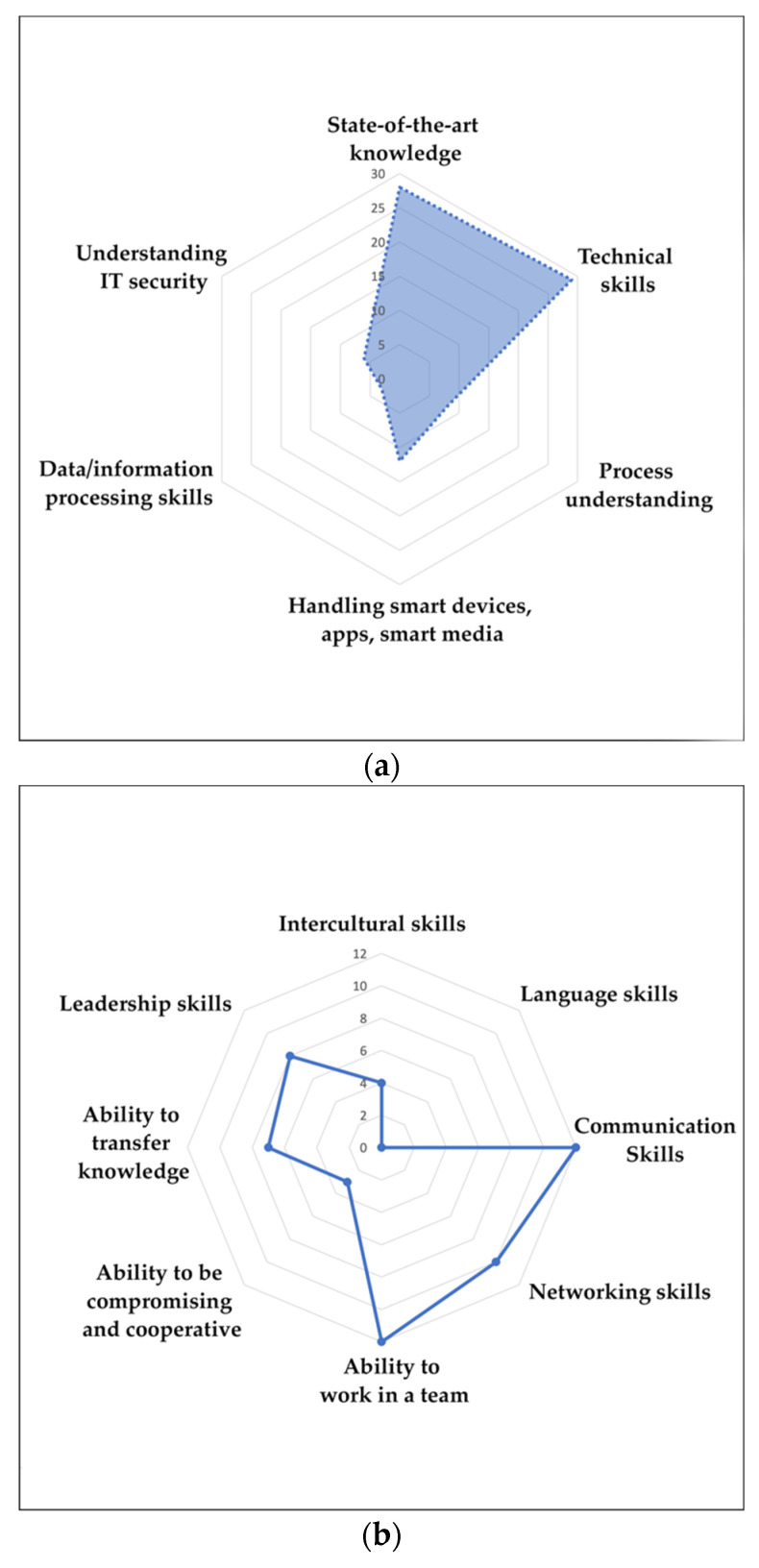
(**a**) Radar chart of technical competence based on the number of citations; (**b**) radar chart of methodological competence based on the number of citations; (**c**) radar chart of social competence based on the number of citations; (**d**) radar chart of personal competence based on the number of citations.

**Table 1 ijerph-20-00478-t001:** Modified competence framework based on Hecklau [9].

Competency Type	Technical Competency	Methodological Competency	Social Competency	Personal Competency
**Definition**	Individual’s abilities to handle different ICTs, large amounts of data, and work virtually with associated sub-skills.	Individual’s abilities to be innovative, involved in strategic tasks, resolve potential issues in the tasks efficiently, and learn continuously with associated sub-skills.	Individual’s ability to work with people from diverse backgrounds, utilise knowledge networks, transfer work and knowledge, and take on more responsibilities.	Individual’s abilities to be flexible with job profiles, accept changes, be ready to learn new things and support sustainable initiatives, manage pressure, and understand regulatory requirements.
**Associated sub-skills**	State-of-the-art knowledge Technical skills Process understanding Handling smart devices, apps, smart media Data/information processing skills Understanding IT security	Creativity Entrepreneurial thinking Problem solving Conflict solving Decision making Analytical skills Research skills Efficiency orientation	Intercultural skills Language skills Communication skills Networking skills Ability to work in a team Ability to be compromising and cooperative Ability to transfer knowledge Leadership skills	Flexibility Ambiguity tolerance Motivation to learn Ability to work under pressure Sustainable mindset Compliance

**Table 2 ijerph-20-00478-t002:** Initial search results of the database.

Keywords	Database	Initial Screening
(Healthcare 4.0 OR Smart health OR eHealth OR mHealth OR Digital health) AND (Hospital) AND (Competenc* OR Skills OR Knowledge)	Web of Science	234
	PubMed	233
	Scopus	264
	Total	731

**Table 3 ijerph-20-00478-t003:** Filtration and selection of studies.

Keywords	Database	Initial Screening	Articles Filtering Steps			
			Removal of Duplicate Articles	Article’s Titles Aligned with the Research Topic	Abstracts Aligned with the Research Topic	Article’s Full Text Aligned with the Research Topic
(Healthcare 4.0 OR Smart health OR eHealth OR mHealth OR Digital health) AND (Hospital OR Clinic OR Nursing home OR Health service OR Care provider) AND (Competenc* OR Skills OR Knowledge OR Expertise OR Literacy OR Learn* OR Experience)	Web of Science	774	4262	209	121	51
	PubMed	804				
	Scopus	3398				
	Total	4976				

**Table 4 ijerph-20-00478-t004:** Publication year vs. count.

Year	Count
2011	1
2012	1
2014	1
2015	2
2016	4
2017	2
2018	3
2019	10
2020	15
2021	12

**Table 5 ijerph-20-00478-t005:** Journal name vs. count.

Journal Name	Count
BMC Health Services Research	3
International Journal of Medical Informatics	3
Health Informatics Journal	2
PLoS ONE	2
Others	41

**Table 6 ijerph-20-00478-t006:** Healthcare 4.0 ICTs, role, and applications.

H4.0 ICTs Sr. No.	Name of ICTs/Role	Non-Clinical Applications	Clinical Application
t1	Biomedical/digital wireless sensors/(S–C)	-	Medical sensors [38] Wireless body area networks [33] Enhanced patient monitoring [39]
t2	Internet of things (IoT)/(S–C)	mHealth/eHealth [40,41,42] Mobile apps [43] Telemedicine/telehealth [44,45,46,47] Electronic prescriptions [48] e-billing systems [49] Barcode scanning for patents [50] Electronic health record/medical health record [51,52] Electronic health-information resource [53] Laboratory information and administration management system [48] Patient portals [54]	Monitoring of patients [55,56]
t3	Big data/(S–C)	-	A large amount of health data, big data analytics for healthcare [33,38]
t4	Cloud/fog computing/(S–C)	Cloud-based medical information systems [33] Cloud technology for professional training [57]	Healthcare data stored in cloud [33]
t5	Remote control/monitoring/(S–C)	Remote consultations [58]/remote medical reporting [46]	Remote assistance [46], remote patient rounding/remote monitoring of patients [56]
t6	Collaborative robots/robotics/(P–A)	-	Machine-assisted procedures [39]
t7	Machine/deep learning/(P–A)	-	Informed decisions [55]

**Table 7 ijerph-20-00478-t007:** Technical competence for adopting H4.0 ICTs and cited literature.

State-of-the-Art Knowledge								Technical Skills					Process Understanding			Handling Smart Devices, Apps, Smart Media	Data/Information Processing Skills	Understanding IT Security
Professional Knowledge of Individual	Knowledge to Handle Various Technologies	Knowledge of System-Specific Tasks	Knowledge Acquired from Experience	Knowledge Acquired from Interactions and Meetings	Knowledge to Prioritise the Needs	Knowledge of the Pre-Operative Planning Process	Knowledge of Data Protection	Technical Skills in the Workplace	Handling of Healthcare 4.0 and Other eHealth Technologies	Operation of Computer Systems and Computer Literacy	Device Testing and Troubleshooting	Evaluation Abilities	Understanding of Related Processes and Procedures	Feedbacks Processed with Data	Hardware and Software Processing			
Kukhtevich [57] Elkefi [35] Rubbio [59] Curtis and Brooks [32] Varsi [60] Houwelingen [61] Gjellebæk [62] Ha and Nuntaboot [63] Buchel t [15] Fernandes [39] Gartrell [54] Marutha [52]	Kiberu [44] Wernhart [64] Schwarwz [65] Kuek and Hakkennes, [66] Saleh [36] Öberg [48] Chen [67] Vanagas [68] Albarrak [45] Herlambang [69]	Jensen and Kushniruk [42] Moss [50] Varsi [60]	Yusif [34] Marutha, [52] Bravo [38] Chen [67]	Yusif [34]	Karahanna [70]	Dyb [71]	Houwelingen [58]	Kukhtevich [57] Rubbio [59] Moss [50] Schwarz [65] Ha and Nuntaboot, [63] Buchelt [15] Fernan des [39]	Kiberu [44] Ogoe [41] Albarrak [45] Curtis and Brooks, [32] Houwelingen [58] Faloye [55] Marutha [52] Kuek and Hakkennes [66] Houwelingen [61] Gardas [33] Chao [72] Durrani [73]	Landis-Lewis [31] Benwell [74] Tesfa [53] Stadin [49] Shiferaw [56] Saleh [36] Öberg [48] Herlamb ang [69]	Jensen and Kushniruk, [42] Loeb [47] Karahanna [70]	Gartrell [54]	Albarrak [45] Loeb [47] Marutha [52] Varsi [60] Ha and Nuntaboot [63] Fernandes [39]	Landis-Lewis [31]	Karahanna [70]	Albarrak [45] Landis-Lewis [31] Jensen and Kushniruk [42] Loeb [47] Curtis and Brooks [32] Houwelingen [58] Tesfa [53] Shiferaw [56] Kuek and Hakkennes [66] Saleh [36] Öberg [48] Moss [50]	Landis-Lewis [31] Marutha [52] Shiferaw [56]	Kamenjasevic and Povese [40] Anyanwu [43] Curtis and Brooks [32] Houwelingen [61] Houw elingen [58] Vanagas [68]

**Table 8 ijerph-20-00478-t008:** Methodological competencies for adopting H4.0 ICTs and cited literature.

Creativity	Entrepreneurial Thinking	Problem Solving	Conflict Solving	Decision Making	Analytical Skills	Research Skills	Efficiency Orientation	Creativity
Anyanwu [43] Karahanna [70] Prameswari [75] Buchelt [15]	Buchelt [15]	Jensen and Kushniru [42] Loeb [47] Rubbio [59] Houwelingen [58] Scharwz [65] Marutha [52] Shiferaw [56] Buchtel [15] Gartrell [54] Houwelingen [58]	Buchelt [15] Gartrell [54]	Jensen and Kushniruk [42] Rubbio [59] Houwelingen [58] Scharwz [65] Houwelingen [61] Buchtel [15] Karahanna [70] Elkefi [35] Yusif [34] Fernandes [39] Gartrell [54] Buchtel [15]	Shiferaw [56] Kuck and Hakkennes [66] Fernandes [39] Gartrell [54] Buchelt [15] Jensen and Kushniruk [42]	Fernandes [39]	Buchelt [15]	Anyanwu [43] Karahanna [70] Prameswari [75] Buchelt [15]

**Table 9 ijerph-20-00478-t009:** Social competencies for adopting H4.0 ICTs and cited literature.

Intercultural Skills	Language Skills	Communication Skills	Networking Skills	Ability to Work in a Team	Ability to Be Compromising	Ability to Transfer Knowledge	Leadership Skills
Jensen and Kushniruk [42] Tortorella [76] Durrani [73] Ha and Nuntaboot [63]	-	Jensen and Kushniruk [42] Yusif [34] Elkefi [35] Loeb [47] Curtis and Brooks [32] Marutha [52] Shiferaw [56] Houwelingen [61] Gjellebæk [62] Varsi [60] Chao [72] Buchtel [15]	Occelli and Scelfo [46] Karahanna [70] Yusif [34] Scharwz [65] Durrani [73] Gjellebæk [62] Varsi [60] Buchelt [15] Fernandes [39] Vanagas [68]	Jensen and Kushniruk [42] Rubbio [59] Dyb [71] Scharwz [65] Marutha [52] Shiferaw [56] Gjellebæk [62] Bravo [38] Varsi [60] Gardas [33] Chao [72] Watterson [77]	Stadin [49] Gardas [33] Watterson [77]	Jensen and Kushniruk [42] Karahanna [70] Tortorella [76] Houwelingen [58] Benwell [74] Scharwz [65] Houwelingen [61]	Scharwz [65] Marutha [52] Gjellebæk [62] Varsi [60] Gardas [33] Prameswari [75] Buchelt [15] Fernandes [39]

**Table 10 ijerph-20-00478-t010:** Personal competencies for adopting H4.0 ICTs and cited literature.

Flexibility	Ambiguity Tolerance	Motivation to Learn	Ability to Work under Pressure	Sustainable Mindset	Compliance
Anyanwu [43] Rubbio [59] Stadin [49] Buchtel [15] Fernandes [39]	Rubbio [59] Scharwz [65] Marutha [52] Varsi [60] Saleh [36] Öberg [48]	Kiberu [44] Elkefi [35] Rubbio [59] Scharwz [65] Varsi [60] Öberg [48] Buchtel [15] Fernandes [39]	Varsi [60] Öberg [48]	Anyanwu [43]	Kamenjasevic and Povse [40] Kiberu [44] Albarrak [45] Jensen and Kushniruk [42] Occelli and Scelfo [46] Tortorella [76] Loeb [47] Houwelingen [61] Ha and Nuntaboot [63] Vanagas [68]

**Table 11 ijerph-20-00478-t011:** Relevance of competence in adopting different ICT forms based on citation frequency.

Competence Type	S–C					P–A	
	t1	t2	t3	t4	t5	t6	t7
Technical	1	2	1	1	1	1	1
Methodological	1	1	0	0	1	1	0
Social	1	2	1	1	1	1	1
Personal	1	2	1	1	1	1	1

**Table 12 ijerph-20-00478-t012:** Evidenced impact level of competence on adopting different ICT forms.

Competence Type	S–C					P–A	
	t1	t2	t3	t4	t5	t6	t7
Technical	1	2	1	1	1	1	1
Methodological	1	2	1	1	1	1	1
Social	1	2	1	1	1	1	1
Personal	1	2	1	1	1	1	1

**Table 13 ijerph-20-00478-t013:** Overall relevance of competence in adopting different ICT forms.

Competence Type	S–C					P–A	
	t1	t2	t3	t4	t5	t6	t7
Technical	2	4	1	1	2	1	1
Methodological	1	4	0	0	1	1	0
Social	1	4	1	1	2	1	1
Personal	1	4	1	1	1	1	1

## Data Availability

Not applicable.

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
