# Peer review of "Professional Competence and Its Effect on the Implementation of Healthcare 4.0 Technologies: Scoping Review and Future Research Directions"

_ijerph, 2022, doi:10.3390/ijerph20010478_

Round 1

Reviewer 1 Report

The authors propose a scoping review of the literature regarding the competences to be possessed by healthcare workers. Even though the present work is presented as a scoping review it follows the methodology suggested by PRISMA statement and, more important, it succeeds in being scientifically sounding (i.e., it is reproducible and questionable). In order to address the interpretative task (the harder one in every literature review) the authors brilliantly deal with the documental corpus by following the classification provided by Hecklau and Tortorella. The main conclusions seem to depict a health care system in which workers are asked to change their competences; nurses are willing to embrace this transition process while physicians seem more reluctant. Finally, the authors proposed a research agenda accordingly to the review’s results which can be summarized in three points:

·        The need for a cross-sectional study to find the ultimate set of competences required for adopting H4.0 ICTs;

·        The chance of building an assessment tool to be used for job placement capable to link a specific competences’ set to a certain job;

·        The room for an evaluation of the possible interactions between the H4.0 and organizational learning capabilities.

In general, the article is well both written and organized, and the underlying research has been conducted rigorously and it does a good service to the international journal of environmental research and public health.

I recommend its publication after solving some minor issues

In particular:

1.    Adjust the page numbering that is compromised (e.g., page 2, 11, 16, 18 of the .pdf file are all numbered as page 2/27, etc.);

2.    Table 4 and table 5 report count per year and per journal name; it would be interesting to join these information reporting count per journal over years (and then summing per journal and per year). As a further improvement one could consider the single journal’s aim and scope to highlight the evolution of the corpus;

3.    As a matter of personal taste, I would prefer to see captions more wordy than those in the article; each table and figure should stand out and even when considered separately from the main text; as a general rule try to lessen the effort required by the reader;

4.    the resilience of a health care organization, cannot be identified with practices such as remote consultation, virtual medical data management, interconnected support for medical emergencies, etc. I think that inadvertently, the authors call into question a concept that is likely to be misleading - that of resilience of health care organizations (I suggest you Braithwaite, Jeffrey, Robert L. Wears, and Erik Hollnagel, eds. Resilient health care, volume 3: reconciling work-as-imagined and work-as-done. CRC Press, 2016. The research domain is quite specific and the misuse of the term “resilient” might be used as leverage for a rebuttal. For example, I can’t see how reducing reliance on human adaptive skills (line 46) could enhance resilient performance of healthcare systems; indeed, Resilience Engineering theory claims that humans are the ultimate mean to resilient performance. Line 51-53 pose an argument supporting the thesis that proneness to brittle performances can be more related to extensively digitalized system, which is against the idea of human in the loop feature of resilient sociotechnical systems. I would rephrase these sentences to eliminate any reference to resilience, since this is not concerning the conducted review;

5.    Among the skills that emerged, the ability to cope with information overload seems to be totally lacking. Since this is a known problem associated with the introduction and increase of new technologies, how do the authors explain this result? Is it a result associated more with organizational issues and simple health care management of complexity (i.e., simplexity in health care), or is it something dependent on the interpretive lens chosen? (i.e., Hecklau and Tortorella taxonomies);

6.    In section 2.3, the authors report that most of the papers discarded at the screening stage were identified only by reading the title; Don't the authors think that this selection filter may be too strict? Some scholars (many, I guess) may have simply chosen the title poorly. Now, this may still be a defensible choice for reasons of practicality (although I think this is the only real flaw in the work), the problem is that this important limitation is not reported in the conclusions

Author Response

We would like to thank reviewer 1 for all of his/her comments which have significantly improved the paper. Attached is a word file with the response for reviewer 1.

Reviewer 2 Report

Formatting issues, authors need to review and ensure that all of the formatting is consistent throughout the manuscript, as well as, with the author instructions.  You need to pay attention to in-text citations, in some locations you have a space before the citation, in others you do not.  Punctuation is an issue throughout as well.

There are several major grammatical errors that need to be addressed. Make sure that the tense (present-tense or past-tense), is consistent throughout the manuscript. It changes in several places.

Needs to be written in ONE voice, it is easy to detect when the author has changed. 

(See attached file for detailed comments)

Author Response

We want to thank reviewer 2 for all of his/her comments, which have significantly improved the paper. Attached is a word file with the response for reviewer 2.

Reviewer 3 Report

This is a very interesting manuscript, which seeks a solid foundation provided by literature written since 2011 on the skills needed for professionals in the health sector, associated with the growing presence of new technology in the work activity of these professionals, specifically technology 4.0. This technology brings the advantage of helping to solve situations without being dependent on a possible overestimation of human capabilities aimed at permanent adaptation to new and unusual situations that occur in the daily lives of health professionals. However, the use of 4.0 technology in the functional contents of these professionals implies a new reconfiguration of the interconnection between soft and hardskills present in these professionals.

What this manuscript proposes is precisely the multidimensional consideration of the reconfiguration of the skill set of health professionals in the face of the unavoidable introduction of technology 4.0 in the health sector, or H4.0 as it is referred to in the text.

A framework of competencies is identified where different types of competencies are highlighted that should be present in the health professional in their professional practice, such as Technical Competency, Methodological Competency, Social Competency, and Personal Competency.

Basically, what is done is to propose the theoretical problematic to be adopted in studies that deal either with the functional content of the different health professions, or with the set of technical, personal and social competencies, saying which are the concepts, dimensions, variables and indicators to be invested in.

The literature review, deep and extensive, also allows us to identify the reference authors and manuscripts to be consulted for the construction of the problematics and theoretical models of analysis in future studies.

The methodology of data analysis and presentation followed is convincing and well explained. The graphical presentation supporting the answers to the research questions is well done, showing the incidence themes of the studies already done and published, present in Scopus, Web of Science and PubMed indexes, relating competencies and technology 4.0. The debugging process is well described, and well linked to the results presented. 

The presentation of radar chart data is interesting, complementing the presentation of data on the investment made in different types of competencies in their association with Industry 4.0 in the health sector.

Very interesting is also the proposal of a future research agenda in areas that have been less invested in scientific studies: Validation of the different types of competence in adopting Healthcare 4.0; Examine the need of job specific competence; Explore the impact of the adoption of H4.0 on organizational learning capabilities.

One remark: in references, number 84 is empty, and should be deleted.

Author Response

We would like to thank reviewer 3 for all of his/her comments which have significantly improved the paper. Attached is a word file with the response for reviewer 3.

Round 2

Reviewer 2 Report

Thank you for your responses to my review.  The manuscript has a smoother flow to it now.